# Novel Treatments against *Mycobacterium tuberculosis* Based on Drug Repurposing

**DOI:** 10.3390/antibiotics9090550

**Published:** 2020-08-28

**Authors:** Álvaro Mourenza, José A. Gil, Luis M. Mateos, Michal Letek

**Affiliations:** 1Departamento de Biología Molecular, Área de Microbiología, Universidad de León, 24071 León, Spain; amouf@unileon.es (Á.M.); jagils@unileon.es (J.A.G.); 2Instituto de Biología Molecular, Genómica y Proteómica (INBIOMIC), Universidad de León, 24071 León, Spain; 3Instituto de Desarrollo Ganadero y Sanidad Animal (INDEGSAL), Universidad de León, 24071 León, Spain

**Keywords:** *Mycobacterium tuberculosis*, multidrug-resistant strains, host-directed therapies, drug repurposing

## Abstract

Tuberculosis is the leading cause of death, worldwide, due to a bacterial pathogen. This respiratory disease is caused by the intracellular pathogen *Mycobacterium tuberculosis* and produces 1.5 million deaths every year. The incidence of tuberculosis has decreased during the last decade, but the emergence of MultiDrug-Resistant (MDR-TB) and Extensively Drug-Resistant (XDR-TB) strains of *M. tuberculosis* is generating a new health alarm. Therefore, the development of novel therapies based on repurposed drugs against MDR-TB and XDR-TB have recently gathered significant interest. Recent evidence, focused on the role of host molecular factors on *M. tuberculosis* intracellular survival, allowed the identification of new host-directed therapies. Interestingly, the mechanism of action of many of these therapies is linked to the activation of autophagy (e.g., nitazoxanide or imatinib) and other well-known molecular pathways such as apoptosis (e.g., cisplatin and calycopterin). Here, we review the latest developments on the identification of novel antimicrobials against tuberculosis (including avermectins, eltrombopag, or fluvastatin), new host-targeting therapies (e.g., corticoids, fosfamatinib or carfilzomib) and the host molecular factors required for a mycobacterial infection that could be promising targets for future drug development.

## 1. Introduction

*Mycobacterium tuberculosis* is the causative agent of tuberculosis (TB) and it is now considered the leading cause of death due to a bacterial infection worldwide [1]. Recent estimates suggest that more than 30% of the world population is infected with *M. tuberculosis* and 10 million people develop the disease every year [1]. Moreover, the incidence of infections caused by multidrug-resistant tuberculosis (MDR-TB) strains (resistant to rifampicin and isoniazid) is rising notably in some parts of the world such as Africa and Asia [2,3]. It is estimated that approximately 500,000 people are infected by MDR strains every year and less than half of treated patients finish the lengthy treatments required for total remission, which leads to high mortality rates [3]. Due to the seasonality of the disease, computational models have been developed in an attempt to better control the incidence of the disease [4].

During infection, *M. tuberculosis* is capable of intracellular replication within alveolar macrophages. These bacteria can modulate the immune response by controlling the maturation of macrophages, which keeps the infection active and drives transmission [1,5]. Moreover, the metabolism of *M. tuberculosis* is adapted to the changing intracellular environment, and the pathogen can control the metabolism of infected macrophages [1,5].

During the initial stages of host cell infection, *M. tuberculosis* block macrophage maturation by inhibiting the fusion between the pathogen-containing vacuole and lysosomes, which is a key process to acidify the intraphagosomal environment and kill bacteria. Besides, *M. tuberculosis* elicits changes in the macrophage’s proteome and glycoproteome, as well as changes in the proteome composition of microparticles secreted by infected macrophages that are important to activate an inflammatory response to the infection [6]. In addition, *M. tuberculosis* profoundly alters the transcriptome of the infected host cells [7,8,9], leading to changes in the innate immune response and the carbon central metabolism, which facilitates the dispersion of the bacteria through the host to secondary points of infection [10,11].

The increasingly detailed knowledge on the host–*M. tuberculosis* interactions has recently promoted the identification and development of host-directed therapies (HDT) [12], which could be used as adjuvant therapies for infections caused by MDR strains. These strategies work in combination with traditional antibiotherapy, and therefore novel antimicrobials are still required. Fortunately, drug repurposing of anti-infectives is presently being considered as a very promising pathway to the identification of novel therapeutic options against MDR-TB, and also extensively drug-resistant tuberculosis (XDR-TB) strains, which are causing untreatable infections [13]. Drug repurposing has the main objective of reducing the first stages of the drug development process. This simplifies the pre-clinical research work and removes the need for lengthy, secure clinical trials, which reduce the time and the investment needed to find new treatments [14,15]. There are several experimental and in-silico approaches to find drugs with repurposing potential, but the most commonly used are knowledge-based approaches, molecular docking, and phenotypic screening [16].

## 2. Repurposing Anti-Infectives against *M. tuberculosis*

The number of new antimicrobials developed against *M. tuberculosis* has not increased significantly over the last few years for a variety of reasons [3]. First, many antimicrobials are ineffective against *M. tuberculosis* in vitro, and those that are effective may not reach the pathogen intracellularly due to poor permeability across host cell membranes. Moreover, some of the first-line antimicrobials are ineffective due to a rapid selection of mutants carrying changes in specific genes that confer resistance to the drug. For example, pyrazinamide resistance in *M. tuberculosis* is due to mutations in the gene coding for pyrazinamidase [17,18]. In addition, the most severely affected countries by tuberculosis are low-income countries, and this disease is not a priority for high-income regions. Hence, the available funding for the research and development of new drugs against tuberculosis is limited when compared to the resources existing for other diseases with a similar death toll, such as diabetes [according to the Estimates of Funding for Various Research, Condition, and Disease Categories (RCDC) of the National Institutes of Health (NIH)]. Finally, this research work is not keeping pace with the fast evolution of multidrug-resistant *M. tuberculosis* strains [3].

During the last 40 years, only a small number of antituberculosis compounds have been approved for clinical use [19]. In addition, some of the most promising drugs, such as bedaquiline (an inhibitor of the mycobacterial ATP synthase), are efficiently excreted by the pathogen through bacterial efflux pumps [19]. Therefore, traditional antibiotherapy could now be complemented with efflux-pump inhibitors, such as verapamil, to increase the efficiency of antimicrobial drugs that target the pathogen [19]. In addition, the small variety of molecular scaffolds discovered so far with anti-infective properties is fuelling the rapid rate at which antimicrobial resistance is rising [20]. Thus, there is active research focused on the search for new molecular scaffolds with anti-infective activity.

At the same time, drug repositioning is now viewed as a very promising therapeutic strategy to reduce the gap between the increase in drug-resistance and the development of new antibiotics [21]. A general approach to the repurposing of antimicrobials is based on broad-spectrum antimicrobial screening for antituberculosis activity with targets that are essential bacterial proteins such as bacterial ribosomal proteins, biofilm formation factors, or proteins involved in general biosynthetic pathways (Figure 1A) [3,22].

Molecular docking is a useful strategy to analyse the interaction of antimicrobials with specific targets. The efficient use of molecular docking techniques requires a profound knowledge of the bacterial proteome in order to identify promising targets. These could be essential proteins of the pathogen or molecular factors involved in host colonization or intracellular survival [15]. This information could be used for in silico screenings with antimicrobials that have already been approved for other purposes. This approach has recently been successful in identifying 20 different compounds with good antimicrobial activity, such as eltrombopag and fluvastatin [15] (Table 1 and Figure 1A).

Other strategies can be based on the analysis of oxidative stress-generating compounds and their combination to increase the biosynthesis of radical oxygen species (ROS) during host cell infection [44,45]. During phagocytosis, *M. tuberculosis* is exposed to oxidative stress, but this is not sufficient to kill the pathogen [46]. However, ROS-generating antimicrobials may increase the efficiency of the free radical biosynthesis produced by macrophages during the oxidative burst, which may facilitate the phagocytosis of the pathogen [46]. Several antituberculosis drugs have been tested as promising ROS-generating antimicrobials against *M. tuberculosis*, and the majority of them have produced an oxidative shift during infection, especially clofazimine [46], but also rifampicin and isoniazid [47]. The combination of several ROS-generating compounds could be a new solution against intracellular *M. tuberculosis* as it was demonstrated for other intracellular pathogens [44,45,48].

In addition, some broad-spectrum anti-helminthic drugs, such as avermectins, have shown promising antimicrobial activity against *M. tuberculosis* in vitro (Table 1 and Figure 1A) [24]. Moreover, the use of transition metals as Cu^2+^ and Co^2+^ associated with benzohydroxamate showed good results against intracellular *M. tuberculosis* (Figure 1A). However, further research is required to understand their mechanisms of action and to discard any cytotoxic effects on human cells [23].

Due to the limited success in repurposing anti-infectives targeting the pathogen, the screening of antitubercular therapies is expanding to drugs that have never been used to treat infections, such as anticancer or antipsychotic drugs [3,24,49]. This research work has already identified very promising antituberculosis compounds. These include eltrombopag and fluvastatin, which have important antimicrobial activities in vitro and during infection, probably due to their inhibition of *M. tuberculosis* Zmp1 and PDF proteins (Table 1) [15].

Moreover, many antipsychotic drugs have shown antimicrobial activity only at high doses, which is accompanied by important side effects. However, some non-neuroleptic derivatives of phenothiazine have shown antimicrobial activity in vitro and in vivo against different pathogens, including *M. tuberculosis*, without causing adverse side effects [25].

## 3. Host-Directed Therapies (HDT) against *M. tuberculosis*

Drug repositioning of host-directed therapies is becoming a very promising approach to find novel combinatorial therapies against many antimicrobial-resistant pathogens. With this objective, the “Host-directed Therapies Network” has been working since 2015 to find novel strategies against *M. tuberculosis* [50]. The main objective of the network is to perform randomized and placebo-controlled clinical trials with HDTs used as adjuncts to traditional antibiotherapy. This network aims at shortening the length of the treatments, improving treatment outcomes, preventing permanent lung damage, and improving the mortality rate of patients with comorbidities such as cancer or cardiac disease [50]. This is important since patients who have coursed tuberculosis often suffer from chronic lung impairment, which is mainly due to an inadequate inflammatory response [3]. Therefore, the fine tuning of the host response could be an important step for preventing the long-lasting effects of tuberculosis and increasing the life expectancy of patients [50,51].

Host-directed therapies could be often targeted to the inhibition of host molecular factors that are important for the intracellular survival of *M. tuberculosis*. However, HDTs may also be used to activate specific antimicrobial routes [3]. Moreover, a strong immune system is usually enough to control *M. tuberculosis* proliferation [29]. Therefore, immunotherapeutics is a very promising approach for the development of novel anti-tuberculosis therapies. Besides, novel treatments based on modulating the host–pathogen interactome are also under development (Figure 1B) [50].

Moreover, some HDTs may be used to treat tuberculosis and other comorbidities at the same time. For instance, metformin is a clinically approved drug used to control type 2 diabetes mellitus that has shown anti-tuberculosis activity (Table 1). In particular, metformin facilitates phagosome–lysosome fusion (Figure 1B) and increases ROS concentration during the oxidative burst, which inhibits the bacterial colonization, reduces lung damage and chronic inflammation, enhances the immune response against tuberculosis and increases the activity of classical anti-tuberculosis drugs [26,29,52]. Indeed, diabetes mellitus patients are more susceptible to bacterial infections due to a depressed immune system. Therefore, metformin may reduce the comorbidity of both diseases and improve the immune system response [26,52].

Similarly, statins are used in the treatment of atherosclerotic cardiovascular disease and hypercholesterolemia [27], and they are another very promising source of antimicrobial compounds against *M. tuberculosis*. For example, simvastatin is capable of reducing the intracellular bacterial load when combined with other antitubercular compounds [24,27]. The mechanism of action of this drug seems to be related to the inhibition of the enzyme 3-hydroxy-3-methylglutaryl-coenzyme A (HMG-CoA) reductase, which may alter the cholesterol levels of the phagosomal membrane (Table 1) and this overcomes the restricted maturation of phagosomes containing *M. tuberculosis* (Figure 1B) [24,27]. Moreover, simvastatin also acts as an activator of cellular immunity by increasing the release of cytokines such as IL-10 [28].

Similarly, corticoids may be employed against tuberculosis to reduce lung pathology during the first stages of the disease (Figure 1B) [29,30]. Other modulators of the host immune system with promising anti-tuberculosis activity include rapamycin, valproic acid, or ibuprofen [33,51]. In addition, doxycycline is a clinically-approved tetracycline that may act as a matrix metalloprotease inhibitor (Table 1), which may reduce tissue damage [3,31,32].

Finally, nitazoxanide (NTZ) is an anti-protozoan compound that has shown antimicrobial activity against *M. tuberculosis* by inducing autophagy at low concentrations (Figure 1B) [51,53]. NTZ augmented the expression of several host factors (Table 1), including retinoic acid-inducible protein I (RIG-I), melanoma differentiation-associated protein 5 (MDA-5), protein kinase R (PKR) and mitochondrial antiviral signalling protein (MAVS), which resulted in the inhibition of the intracellular proliferation of *M. tuberculosis* [33,34]. Moreover, pre-treatment with NTZ showed promising results in preventing *M. tuberculosis*’ colonization of the host [33].

### 3.1. HDTs Based on the Induction of Autophagy and Phagosome Maturation

Autophagy is a lysosome-dependent degradation pathway that is essential for maintaining cellular homeostasis [54,55]. In addition, autophagy could also be used by mammalian cells to kill pathogens and it is considered part of the innate immune response.

Interestingly, the basal levels of autophagy in cells are increased during intracellular infection of *M. tuberculosis* [37,54]. However, this pathogen can reduce the autophagic flux during host cell infection [56]. Therefore, the use of molecules that could trigger autophagy is considered an interesting solution to clear *M. tuberculosis*, even when the pathogen is in a dormant and antibiotic-resistant state [55].

In fact, *M. tuberculosis* possesses different mechanisms to evade autophagy and phagosome maturation that allow these bacteria to survive intracellularly [56,57,58]. Several mycobacterial mechanisms are involved in circumventing autophagy, including lipid virulence factors such as sulfoglycolipids and phthiocerol dimycocerosates that directly inhibit autophagy [59].

Moreover, recent studies revealed that some host microRNAs that are expressed during the immune response (e.g., miR-18a) could also be relevant for the intracellular colonization of *M. tuberculosis* by reducing the expression of LC3, an essential protein for autophagosome biogenesis [60].

Fortunately, several drugs can reactivate the autophagy pathway. One of the best-studied is rapamycin, an immunosuppressive drug that inhibits mTOR, a protein kinase that is considered the central activator of autophagy (Figure 1B). Therefore, rapamycin promotes autophagy and it would be an ideal candidate for the development of novel HDT-based strategies against *M. tuberculosis*, but it is not well absorbed [37,61]. Because of that, other autophagy activators have been tested against this pathogen, which include compounds that activate autophagy as a secondary effect, such as imatinib, metformin, or nitazoxanide. In addition, bazedoxifene is a selective estrogen receptor modulator that inhibits the intracellular proliferation of *M. tuberculosis* by enhancing autophagy in infected macrophages [35]. Moreover, ibrutinib is currently employed against chronic lymphocytic leukaemia, but it also stimulates the expression of LC3 in infected macrophages [36]. Statins increase autophagic flux and control phagosome maturation [62]. Gefitinib, an epidermal growth factor receptor (EGFR) inhibitor is also a HDT candidate against *M. tuberculosis* because it may be an autophagy inductor (Figure 1B and Table 1) [37], although its main mechanism of action in the control of *M. tuberculosis* is due to an activation of the pathogen-containing vacuole trafficking towards lysosomes [63]. Vitamin D regulates inflammatory host responses and activates autophagy (Figure 1B and Table 1), which is why it has been proposed as beneficial against some intracellular pathogens such as *M. tuberculosis* [37,38,39].

### 3.2. Host Genes as Targets for HDTs

During *M. tuberculosis*’ host cell infection, several host factors play crucial roles for the pathogen colonization of the intracellular niche [64]. Novel studies have revealed that changes in the host expression profile could reveal new target genes for novel HDT-based therapies [9]. Interestingly, there are differences in the host gene expression when patients showing active and latent tuberculosis are compared, which suggests that active tuberculosis elicits the expression of different host genes that could be essential for the host cell infection [7,9]. Overall, approximately 90 human pathways have significantly changed in their expression profile during infections caused by *M. tuberculosis* [9]. Moreover, many of the differentially expressed genes are close to tuberculosis-related Single Nucleotide Polymorphisms (SNPs). Therefore, these genes are very attractive targets for the identification of repurposed HDTs. An analysis of 19 of those genes with the “DrugBank” database (https://www.drugbank.ca/) identified some anti-tuberculosis drugs that could control the expression of these targets [9]. This analysis identified carfilzomib, a drug recommended for multiple myeloma, as a promising new drug that could be used to fight *M. tuberculosis* because it is an inhibitor of many of the overexpressed genes during tuberculosis infection. This includes multiple proteasome components such as PSMB8 and PSMB9 [9], but the mechanism of action of this drug during *M. tuberculosis* infections is still unclear [65].

Similarly, type I interferon (IFN) is important for host defence against viral, bacterial, and fungal pathogens [66], although high concentrations of IFN could be detrimental for macrophage activity and may even promote bacterial infections [66]. Nevertheless, IFN-stimulated genes (ISGs) are also very promising targets for drug repurposing since many of these genes are induced during tuberculosis, such as the gene coding for the myxovirus resistance protein 1 (MxA) [67]. The silencing of MxA reduces the infectivity of *M. tuberculosis* by increasing the expression of human cytokines through the activation of the TAK1-IKKα/β-NF-kB pathway [67]. Interestingly, there are microRNAs that may also work as inhibitors of the expression of MxA (Figure 1B and Table 1), which could be an interesting alternative pathway for the development of novel HDTs [40]. However, MxA silencing may facilitate infections caused by the Influenza A virus [17].

The success of the *M. tuberculosis* infection is linked to the generation of microdisruptions in the macrophage membrane [5,56]. These microdisruptions are repaired by prostaglandin E_2_ (PGE_2_), whose expression is blocked by lipoxin A_4_ (LXA_4_) [56,68]. Because of this, the induction of LXA_4_ is a key process controlled by *M. tuberculosis* during infection. Consequently, the PGE_2_ silencing is related to the dispersion of the bacteria and the progression of the infection [56]. Therefore, the genes controlling the expression of PGE_2_ are considered important targets for the development of new antimycobacterial drugs. However, PGE_2_ shows other important biological functions, for example in hematopoietic stem cells’ homeostasis [69]. Therefore, the LXA_4_/PGE_2_ balance should be carefully controlled to disrupt mycobacterial pathogenesis [68].

Moreover, Abl kinases prompt lysosomal function and phagosome maturation, and therefore they could also be good targets for the development of novel therapeutic strategies against *M. tuberculosis* [54].

In addition, apoptosis plays a crucial role in the host defence against intracellular pathogens such as *M. tuberculosis* by preventing the release of the intracellular bacteria. However, virulent strains of *M. tuberculosis* inhibit the apoptotic pathway of infected cells by upregulating the expression of the antiapoptotic *MCL1* gene [70,71]. Some anti-cancer compounds could act as pro-apoptotic drugs, and therefore they could potentially be repurposed against *M. tuberculosis*. For example, cisplatin is an anti-cancerous drug that has been employed against *M. tuberculosis* in vivo due to its proapoptotic and antitubercular activity (Figure 1B and Table 1) [41]. Other promising anti-cancerous and proapoptotic compounds that could be used as antitubercular compounds are calycopterin [42] and different analogs of troxipide [72] (Figure 1B and Table 1).

Similarly, the leucine-rich repeat kinase 2 (LRRK2) seems to be important for the progression of infections caused by *M. tuberculosis* [9]. Indeed, LRRK2-knockout macrophages are able to control *M. tuberculosis* infections, whereas LRRK2-overexpression is essential for the colonization of the host cell by the pathogen [73,74]. However, elevated activity of LRRK2 is also related to sporadic forms of Parkinson’s disease [9,75,76]. Fortunately, fostamatinib could be used to inhibit LRRK2 expression (Table 1). This drug is a spleen tyrosine kinase (SYK) inhibitor approved for the treatment of rheumatoid arthritis and immune thrombocytopenic purpura and has also been granted orphan drug status [43]. In addition, the activation of SYK is related to several haematological cancers [77]. Therefore, the inhibition of the expression of LRRK2 mediated by fostamatinib could be a very promising HDT against tuberculosis.

## 4. General Limitations of Drug Repurposing

It is becoming clear that there are very promising therapeutic strategies against tuberculosis that could be developed by drug repurposing, but there are also some important limitations with this strategy. First, the number of FDA-approved drugs is limited and drug resistance may quickly arise. In addition, drug repurposing still requires validation through biological in vitro or preclinical research, which may reduce the speed of the response against MDR bacteria [16]. Moreover, some of the repurposed drugs may have low activity in vivo and their MICs could be over the maximum dose [16]. Furthermore, the use of pathogen-directed therapies can affect the human microbiome and these antibacterial compounds could induce resistance against antibiotics in other bacteria during the lengthy treatments required to cure tuberculosis patients [78]. Finally, little is known about the resistance that can be generated against host-targeted therapies because of the low number of HDTs approved for their use to treat infections. However, current *M. tuberculosis* strains may become resistant to HDTs, and moreover, it is also possible that other microorganisms become resistant to these new repurposed therapies.

## 5. Conclusions

New tuberculosis treatments are urgently needed to cope with the worrying increase in the incidence of infections caused by antimicrobial-resistant strains. This leads to a higher mortality rate in those patients that course these infections. In addition, new antimicrobials are required to decrease the side effects of tuberculosis, such as permanent lung damage. Drug repurposing is a novel strategy that is gaining interest, because it may facilitate the finding of new and efficient treatments and significantly reduce the necessary time to have new antitubercular drugs in the market [79]. Thanks to this approach, some compounds have already been approved or are under the last stages of clinical research for their use as antitubercular drugs. The most promising drugs are focused on the activation or inhibition of host genes that allow bacterial colonization, which may also lead to a reduction in the selection of new antimicrobial-resistant strains.

## Figures and Tables

**Figure 1 antibiotics-09-00550-f001:**
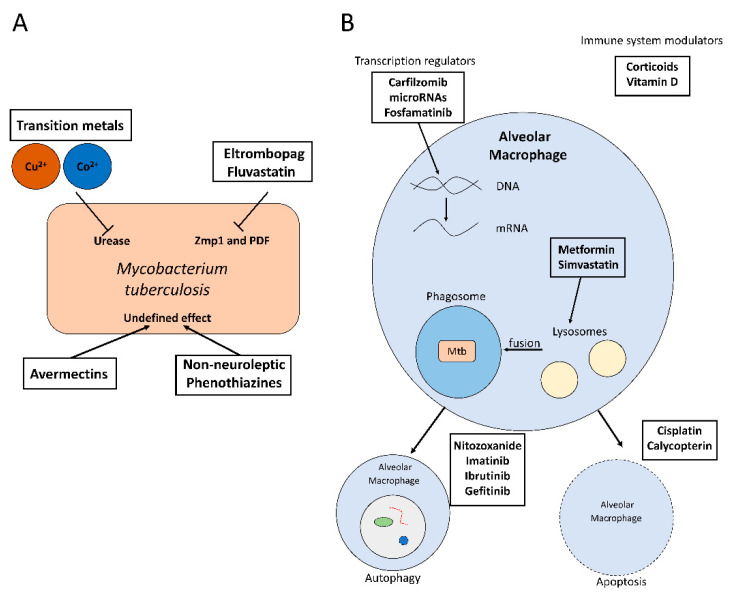
(**A**) Activity of some repurposed antimicrobials against *M. tuberculosis*. (**B**) Mechanism of action of some host-directed therapies to inhibit intracellular *M. tuberculosis* proliferation.

**Table 1 antibiotics-09-00550-t001:** List of antimicrobials that have been repurposed against *M. tuberculosis* (drug repurposing) or focused on the host (Host-directed therapies), and their primary mechanism of action. Zmp1, Zinc-dependent metalloprotease; PDF, peptide deformylase; HMG-CoA, 3-hydroxy-3-methylglutaril-coenzyme A; EGFR, epidermal growth factor receptor; MxA, myxovirus resistance protein 1; LRRK2, leucine-rich repeat kinase 2.

Repurposed Drugs	Primary Mechanism of Action	Reference
Transition metals (Cu^2+^ and Co^2+^)	Interfering with urease	[23]
Eltrombopag Fluvastatin	Inhibition of Zmp1 and PDF	[15]
Avermectin	Undefined	[24]
Non-neuroleptic phenothiazines	Undefined	[25]
**Host-directed therapies**	**Primary Mechanism of Action**	**Reference**
Metformin	Phagosome–lysosome fusion	[26]
Simvastatin	HMG-CoA inhibition	[24,27,28]
Corticoids	Immune system modulation	[29,30]
Doxycycline	Matrix metalloprotease inhibition	[31,32]
Nitazoxanide	Activator of defense host genes	[33,34]
Imatinib Ibrutinib	Autophagy activation	[35,36]
Gefitinib	EGFR inhibition	[37]
Vitamin D	Inflammatory host response regulation	[37,38,39]
Carfilzomib	Host genes inhibition	[9]
microRNAs	MxA inhibition	[40]
Cisplatin Calycopterin	Apoptosis activation	[41,42]
Fostamatinib	LRRK2 and spleen tyrosine kinase inhibition	[43]

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
