# Peer review of "Novel Treatments against Mycobacterium tuberculosis Based on Drug Repurposing"

_antibiotics, 2020, doi:10.3390/antibiotics9090550_

Round 1
Reviewer 1 Report
This is a nice and comprehensive review of drug re-purposing strategies for TB. Re-purposing of existing drugs is for sure a way to address challenges like economics, clinical studies, and drug availability.
The review is comprehensive, but I recommend that the different aspects are commented a bit more critical. E.g. you can re-purpose brought range antibiotics for TB, but this leads all kinds of different problems and would significantly contribute to resistance development (not just against TB but other bacterial strains as well). There is a reason why TB treatment is on the very narrow spectrum.
I also recommend to include some information on how the re-purposed drug are/will be tested.
Otherwise - a very good and interesting article.
Author Response
Many thanks for your kind comments, we have added section 4 (current lines 276-289) to highlight the limitations of the drug repurposing approach and to make emphasis on the need for validation of these therapeutic strategies through biological in vitro or preclinical research. We hope that we have addressed all of the referee's concerns.
Reviewer 2 Report
Reviewer Comments
The review entitled, “Novel treatments against Mycobacterium tuberculosis based on drug repurposing.” The study highlighted the novel antimicrobials against tuberculosis.
The study is a good initiative but the manuscript could not be recommended for acceptance in the current
Abstract:
Line 12, Abstract; [Tuberculosis is the leading cause of death worldwide due to a bacterial pathogen] is needed to be modified as HIV is the leading cause of death. (https://en.wikipedia.org/wiki/List_of_causes_of_death_by_rate)
The word in line 16, “M. tuberculosis” has not been defined earlier.
In the abstract, some new therapies name may be included for reader interest.
Introduction
The word “M. Tuberculosis” in line 31 should be defined first time and then it may be used in the whole manuscript.
Table 1. the word “PDF” has not been defined. Its abbreviated form should be defined at footnote. Similarly all similar words should defined when first come in the text and then abbreviated form may be used.
Line 225, “silencing MxA, reduce infectivity”. The author did not mentioned nor discussed the consequences as MxA inhibit the human influenza A virus replication in primate cells (https://jvi.asm.org/content/87/2/1150) (https://www.ncbi.nlm.nih.gov/pmc/articles/PMC3973384/). Further, the authors mentioned in line 219 and 220, that [carfilzomib, a drug recommended for multiple myeloma, as a promising new drug that could be used to fight M. tuberculosis because it is an inhibitor of many of the overexpressed genes during tuberculosis]. This statement should be elaborated as in myeloma, the targeted therapy may be useful but in case of Mycobacterium infection, the therapies may not be useful instead will have an adverse effect. The myeloma and TB are two different types of diseases.
Line 223, “IFN are very promising targets”, The author should discuss the what kind of adverse effect may occurs while targeting the INF. See this article as an example (https://clincancerres.aacrjournals.org/content/17/9/2619).
Overall comments
My major concern is about the targeting of host genes. As this is a review article, the authors should discuss the main role of host gene before suggesting them as host directed therapy during Mycobacterium infection. Similarly, cancers and mycobacterium infection are two different types of disease. How can a cancer promoting gene may be targeted in mycobacterium infection? Is a totally different scenario. The authors should also highlight the role of host genes before suggesting them as potent targets in TB infection.
Some references should be included:
Mechanistic analysis of A46V, H57Y, and D129N in pyrazinamidase associated with pyrazinamide resistance, July 2020Saudi Journal of Biological Sciences, DOI: 10.1016/j.sjbs.2020.07.015;
Structural Dynamics Behind Clinical Mutants of PncA-Asp12Ala, Pro54Leu, and His57Pro of Mycobacterium tuberculosis Associated With Pyrazinamide Resistance
December 2019Frontiers in Bioengineering and Biotechnology 7:404
DOI: 10.3389/fbioe.2019.00404;
Liao, Z., Zhang, X., Zhang, Y. et al. Seasonality and Trend Forecasting of Tuberculosis Incidence in Chongqing, China. Interdiscip Sci Comput Life Sci 11, 77–85 (2019). https://doi.org/10.1007/s12539-019-00318-x
Author Response
Many thanks for your comments, please find below our answers to all of the points raised during this revision:
Line 12, Abstract; [Tuberculosis is the leading cause of death worldwide due to a bacterial pathogen] is needed to be modified as HIV is the leading cause of death. (https://en.wikipedia.org/wiki/List_of_causes_of_death_by_rate)
Sorry but there must be here some confusion, tuberculosis is the leading cause of death worldwide due to a bacterial pathogen, HIV is a virus.
The word in line 16, “M. tuberculosis” has not been defined earlier.
It is defined in line 13.
In the abstract, some new therapies name may be included for reader interest.
A few examples of some of the therapies covered in this manuscript have been added to the abstract.
The word “M. Tuberculosis” in line 31 should be defined first time and then it may be used in the whole manuscript.
It is defined in line 30.
Table 1. the word “PDF” has not been defined. Its abbreviated form should be defined at footnote. Similarly all similar words should defined when first come in the text and then abbreviated form may be used.
It is now defined in lines 117-121.
Line 225, “silencing MxA, reduce infectivity”. The author did not mentioned nor discussed the consequences as MxA inhibit the human influenza A virus replication in primate cells (https://jvi.asm.org/content/87/2/1150) (https://www.ncbi.nlm.nih.gov/pmc/articles/PMC3973384/).
This point has been mentioned in current lines 243-244.
Further, the authors mentioned in line 219 and 220, that [carfilzomib, a drug recommended for multiple myeloma, as a promising new drug that could be used to fight M. tuberculosis because it is an inhibitor of many of the overexpressed genes during tuberculosis]. This statement should be elaborated as in myeloma, the targeted therapy may be useful but in case of Mycobacterium infection, the therapies may not be useful instead will have an adverse effect. The myeloma and TB are two different types of diseases.
We have added some information on the specific mechanism of action of this drug on tuberculosis infections (current lines 232-234). We have also added Section 4 which is focused on the limitations of drug repurposing. Indeed, some of the drugs may be promising in vitro but have adverse effects in vivo. This depends on the required dose, the secondary effects in patients with tuberculosis, etc. Further research is needed, but one drug could have applications for very different diseases as we have seen recently for COVID-19.
Line 223, “IFN are very promising targets”, The author should discuss the what kind of adverse effect may occurs while targeting the INF. See this article as an example (https://clincancerres.aacrjournals.org/content/17/9/2619).
This is now covered in line 236.
Overall comments
My major concern is about the targeting of host genes. As this is a review article, the authors should discuss the main role of host gene before suggesting them as host directed therapy during Mycobacterium infection. Similarly, cancers and mycobacterium infection are two different types of disease. How can a cancer promoting gene may be targeted in mycobacterium infection? Is a totally different scenario. The authors should also highlight the role of host genes before suggesting them as potent targets in TB infection.
Thanks for this comment, we have included information on the possible function during tuberculosis of all host genes that we have covered in this review. We are trying to emphasize that genes important for cancer disease may also have a role in TB infection and their control could be a promising avenue for the development of novel adjuvant therapies in combination with traditional anti-tubercular compounds. This is the same strategy that has been recently applied to find novel ways to treat COVID-19 patients.
Some references should be included:
Mechanistic analysis of A46V, H57Y, and D129N in pyrazinamidase associated with pyrazinamide resistance, July 2020 Saudi Journal of Biological Sciences, DOI: 10.1016/j.sjbs.2020.07.015;
Structural Dynamics Behind Clinical Mutants of PncA-Asp12Ala, Pro54Leu, and His57Pro of Mycobacterium tuberculosis Associated With Pyrazinamide Resistance December 2019 Frontiers in Bioengineering and Biotechnology 7:404 DOI: 10.3389/fbioe.2019.00404;
Liao, Z., Zhang, X., Zhang, Y. et al. Seasonality and Trend Forecasting of Tuberculosis Incidence in Chongqing, China. Interdiscip Sci Comput Life Sci 11, 77–85 (2019). https://doi.org/10.1007/s12539-019-00318-x
Thanks for your suggestion, we have added those references.
Round 2
Reviewer 2 Report
The authors make significant changes, it is publishable in its current form.